# Understanding water transport through graphene-based nanochannels via experimental control of slip length

Xinyue Wen [1], Tobias Foller [1] ✉, Xiaoheng Jin[1], Tiziana Musso [1], Priyank Kumar [2] ✉ & Rakesh Joshi [1] ✉

The water transport along graphene-based nanochannels has gained significant interest. However, experimental access to the influence of defects and impurities on transport poses a critical knowledge gap. Here, we investigate the water transport of cation intercalated graphene oxide membranes. The cations act as water-attracting impurities on the channel walls. Via water transport experiments, we show that the slip length of the nanochannels decay exponentially with the hydrated diameter of the intercalated cations, confirming that water transport is governed by the interaction between water molecules and the impurities on the channel wall. The exponential decay of slip length approximates non-slip conditions. This offers experimental support for the use of the Hagen-Poiseuille equation in graphene-based nanochannels, which was previously only confirmed by simulations. Our study gives valuable feedback to theoretical predictions of the water transport along graphene-based channels with water-attracting impurities.

Ultrafast water transport through graphene oxide membranes (GOMs) is considered to originate from the friction-free movement of water molecules through sub-nanometre graphene-based channels[1,2]. In detail, GOM nanochannels consist of patches of graphitic and functionalized areas. Along the graphitic parts the water transport is considered frictionless and along the functionalized areas the water transport is partially hindered[3,4]. The assumption that the water transport is friction-less along graphitic domains originates from the use of the Hagen-Poiseuille (HP) equation to describe the water transport in graphene-based nanochannels. The HP equation allows calculating water flux through a channel if water molecules have zero velocity at the liquid/channel wall interface[5]. This condition is called a non-slip condition. In experiments, the liquids often exhibit slip at the channel walls (i.e. the non-zero velocity at liquid/channel wall interface)[5]. To describe the amount of slip, the slip length is introduced. In particular, the water flux through graphene-based nanochannels showed a large slip length. Hence, it was concluded that the water flux is frictionless. In the case of GOMs, the water transport is only partially frictionless due to

patches of functionalized areas. Nevertheless, the GOMs typically show high slip lengths indicating low friction transport along their nanochannels[2]. Recent advances allow a deeper understanding of the unusual water transport along graphene-based planes in nanoconfinement[6–8]. However, it was identified that the influence of defects on this transport is yet challenging to access experimentally[9]. Ab initio molecular dynamic (AIMD) simulations predict that defects in the graphitic plane will cause extra friction to the movement of water inside the nanochannels[10]. The effect on the friction is predicted to increase with the number of hydrogen bonds between the water molecules and the defects[10]. While simulation studies offer a great understanding of the underlying transport mechanism[11], an experimental observation is still lacking. Experimental studies so far mainly focus on how the ionic transport occurs in these nanochannels[3]. However, a systematic study on the water transport in graphene-based nanochannels with water attraction impurities has remained elusive. Thus, controlling the affinity of water molecules towards the nanochannel walls by intercalation of cations with different hydrated diameter may offer a unique

[1]School of Materials Science and Engineering, University of New South Wales, Sydney, NSW 2052, Australia. [2]School of Chemical Engineering, University of New South Wales, Sydney, NSW 2052, Australia. ✉e-mail: t.foller@unsw.edu.au; priyank.kumar@unsw.edu.au; r.joshi@unsw.edu.au

platform to tailor and understand the water friction of the unusual water transport in graphene-based nanochannels.

From a potential application point of view, it is important to understand the water transport in graphene-based nanochannels with impurities. For example, GOMs exhibit promising membrane properties[12,13]. The two key parameters for using GOMs as water purification membranes are first, their capability to reject certain solutes and, secondly, letting pass as much solvent as possible[12]. The recent advance in controlling the interlayer space of GOMs via intercalation of cations is a promising approach to tune the porosity[14,15]. With that potentially controlling the rejection of dissolved ions in aqueous solution using GOMs[16]. Here, the intercalation of $K^+$ prevents swelling of GOMs in a wet environment[16]. Without control, swelling leads to an increase in interlayer space and hence an increase in pore size of the GOMs. Intercalation with $K^+$ prevents swelling and reduces the interlayer space of wet GOMs, allowing the rejection of smaller species[16]. Higher cation concentration led to even lower interlayer space of wet GOMs[17]. The cation intercalated GOMs exhibit various performance improvements, including long-term aqueous stability and strength, and excellent molecular separation performance[15,18].

Previous studies suggest that the water flux is decreased upon intercalation of cations[19,20]. However, a systematic study on how the intercalation of different cations influences the water transport through the nanocapillaries is still lacking. With that, one of the two key parameters in cation intercalated GOMs remains unexplored: the water permeance. This is not only important for potential water purification applications but also for other applications where water transport in GO and graphitic nanochannels with intercalated cations or other impurities displayed high potential[21–23].

In this study, we investigate the influence of cation intercalation on the water transport through GOMs. For that, we intercalate different types of ions (using NaCl, KCl, $MgCl_2$, $CaCl_2$ and $FeCl_3$) with varying water affinities allowing to manipulate the number of hydrogen bonds attracted to the intercalated cation. This leads to the observation of an unintuitive trend. Water flux is decreasing with increasing cation-controlled interlayer space. Via water flux measurements and AIMD simulations, we show that both the water flux and cation-controlled interlayer space are directly correlated to the size of the hydration shell of the respective cation. The use of cations as controlled impurities with different water affinities allows us to manipulate the number of hydrogen bonds attracted to impurities on the wall of a graphene-based nanochannel and experimentally observe its effect on the water transport through graphene-based nanochannels. Further, we gain important insights into the water transport mechanism in graphene-based nanochannels in general and support the use of HP equation to describe the water transport in graphene-based nanochannels.

## Results

### Preparation of cation intercalation GOMs

For the intercalation of cations into the GOMs, two different preparation methods (membrane intercalation and solution intercalation) are common and gave promising rejection results in previous studies[24,25]. To exclude the influence of the different methods, in this study, both methods are used to prepare cation intercalated GOMs. For that, cation intercalated GOMs were fabricated as follows. In the first method (membrane intercalation[16]), the GOMs were prepared via vacuum filtration (Fig. 1a) by filtering GO solution through a porous polymer substrate. Then these membranes were soaked in various salt solutions (Fig. 1b). The membranes from this approach are labelled as X-M-GO, where X stands for the corresponding cation of the salt solution and M indicates that the membrane was intercalated by soaking the GOMs. In the second approach (solution intercalation[23]), the GO solution was mixed with the respective salt solution prior to fabricating the membranes (Fig. 1c). Then the salt-GO mixture was filtered by vacuum filtration through a porous polymer membrane to

obtain the membrane. The membranes prepared from this approach are labelled as X-S-GO. Here, S indicates that the intercalation of GO was initiated in the GO solution by mixing with respective cation X. According to previous studies, the mass increase of dry, cation intercalated GOMs was equal to the mass of intercalated salts[16]. Therefore, the number of moles of cations inside the X-M-GO was calculated from the mass gain of the membranes after soaking the membranes in the respective salt solution. Table 1 shows that the number of moles of cations inside the X-M-GO range from 2 to 7 μmol. The number of moles of cations in the X-S-GO was controlled to match the X-M-GO by adding same number of moles of respective cations into GO solution according to Table 1. In X-M-GO the concentration of intercalated cation is not precisely controllable. To exclude any influence of concentration differences on the water transport, a third group of samples was prepared. Here, the number of moles of intercalated cations was attempted to be constant by mixing appropriate amounts of salt (0.2 μmol) with GO solution (5 mL of 0.1 mg/mL). The molarity of salt was kept low to avoid $Fe^{3+}$ induced aggregation in the mixture. To evaluate the effective intercalated cations, the effluent of the X-S-GO preparation was analysed by ICP-OES for cation concentration. Our results suggest that for different cations, the percentage of intercalation may vary independent of its hydration shell. It may be noted that this experimental approach of testing the effective intercalated cations in such small quantities may not fully represent the actual mass loading. Figure 1d–h presents the structural and chemical characteristics of GOM. The lateral size of GO flakes utilised in this study is ~0.5 μm as displayed in the TEM image in Fig. 1d. The thickness of the GOM was ~200 nm according to the cross-sectional SEM analysis (Fig. 1e). The interlayer space of the dry, original GOM is ~9.4 Å. After soaking in water for 10 mins, the interlayer space in the original GOM increases to 12.3 Å. The interlayer space is depicted via X-ray diffraction (XRD) analysis as described in the methods section.

### Water transport measurements through cation intercalated GOMs

The water transport through GOMs was tested via vacuum filtration. Figure 2a shows the water flux values for membranes prepared using two methods. It becomes clear that for both preparation methods, the water flux decreases following the trend $Mg^{2+}$-M/S-GO < $Ca^{2+}$-M/S-GO < $Na^+$-M/S-GO < $K^+$-M/S-GO. It is thus rational to plot the water flux against the hydrated diameter ($D_H$) of the respective cations as they follow the exact opposite trend of $D_{H, Mg} > D_{H, Ca} > D_{H, Na} > D_{H, K}$. It may be noted that as discussed in the next section the ionic diameters of the ions do not rationally resembles the water flux trend.

As shown in Fig. 2b, the interlayer space increases with the hydrated diameter of the intercalated cations. As previously reported, both methods allow controlling the swelling of GO to a certain degree[16]. In particular, $K^+$-M/S-GO show a smaller interlayer space compared to pristine GOM as well as $Na^+$-M/S-GO. Intercalation of divalent cations ($Ca^{2+}$ and $Mg^{2+}$) increases the wet interlayer spaces in GOMs regardless of the preparation method. In the dry state, X-S-GO show a reduced interlayer space, whereas X-M-GO show an increased interlayer space except for $K^+$. The slight differences in the interlayer spaces resulting from different preparation methods are discussed in Supplementary Notes 1 and Supplementary Fig. 1.

Concentration differences of intercalated cations may be responsible for the observed water flux trend. The number of moles of intercalated cations in the X-M-GO cannot be precisely controlled. However, in the X-S-GO, it is more probable to do so. Therefore, water filtration experiments were conducted on X-S-GO with controlled intercalated cations (0.2 μmol) using $K^+$, $Na^+$, $Ca^{2+}$, $Mg^{2+}$ and $Fe^{3+}$. As shown in Fig. 2c, we find a similar trend as in Fig. 2a that water flux decreased with increasing hydrated diameter of cations. As mentioned above, the number of intercalated cations may also vary for X-S-GO. However, we could not find a matching trend of mass loading and

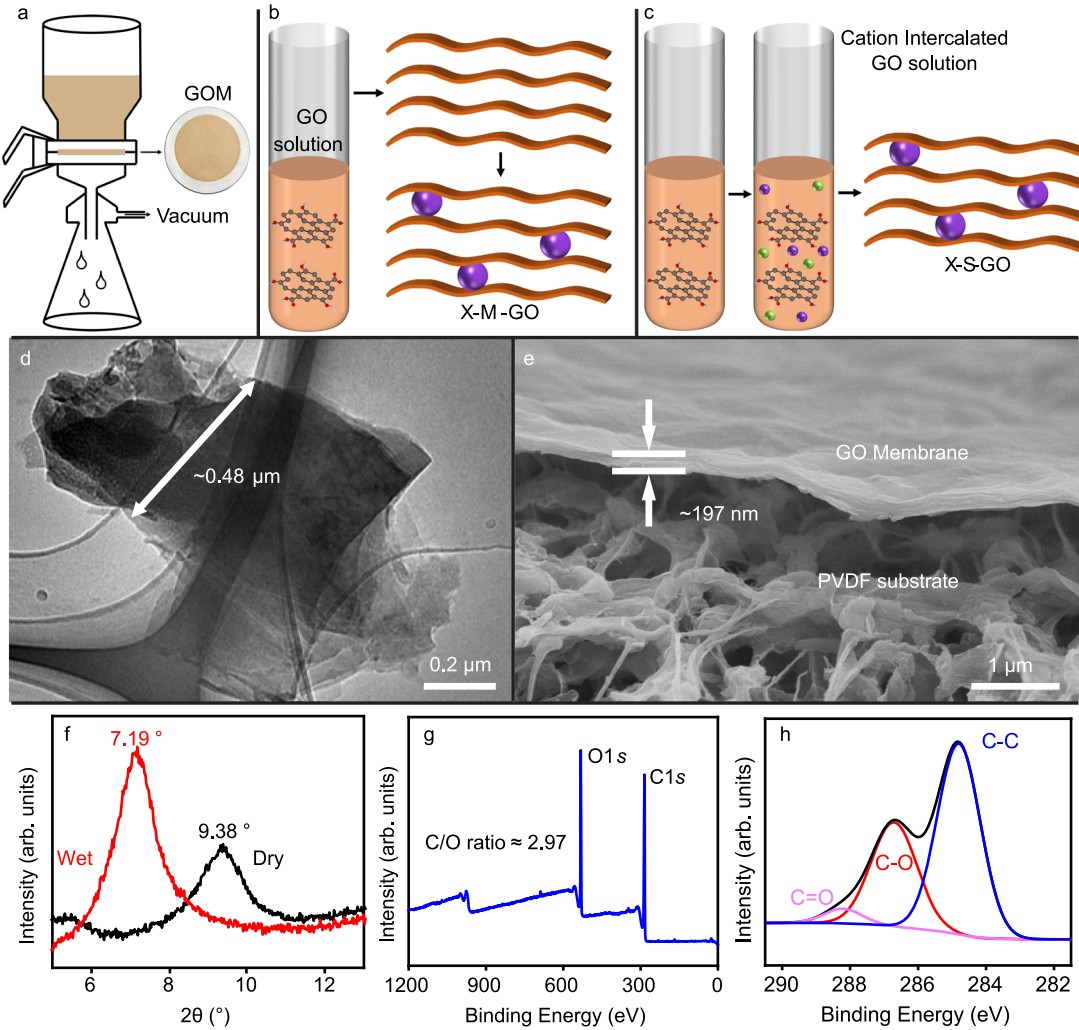

**Fig. 1 | Fabrication and characterisation of graphene oxide. a** Schematic of vacuum filtration setup for graphene oxide membrane (GOM) preparation. **b**, **c** GOM prepared by (**b**) membrane intercalation method (X-M-GO) and (**c**) solution intercalation method (X-S-GO). Violet and green spheres represent the cations and anions, respectively. **d** TEM image of GO sheets showing the lateral size of ~0.48 μm. **e** Cross-sectional SEM image of GOM with thickness ~197 nm on PVDF substrate. **f** XRD pattern of wet GOM (red spectrum, 2θ = 7.19°) and dry GOM (black spectrum, 2θ = 9.38°). **g** XPS survey scan showing O1s and C1s peak. The carbon/oxygen ratio (C/O) is ~2.97. **h** XPS C1s spectra (black curve) showing C=O bond (pink curve) at ~288.1 eV, C−O bond (red curve) at ~286.7 eV and C−C bond (blue curve) at ~284.8 eV. Source data are provided as a Source Data file.

hydrated diameter in any of the experiments. Figure 2a shows that the water flux decreases with increasing hydration diameter despite arbitrary mass loadings of different cations (Table 1). This suggests that the mass loading is not the primary explanation for the observed trend in water flux.

The water flux for membranes prepared using both preparation methods decrease with increasing interlayer space. This is surprising as a decreased interlayer space should offer less room for the water molecules to travel. As reported in our previous study and by others,

this is true for untreated GO where smaller interlayer space resulted in a decline in water flux[20,24]. It may be further noted that all cation intercalated membranes in this study show a decreased water flux in comparison with untreated GO. Combined with the observation that the water flux declines not only with increasing interlayer space but also with increasing hydrated diameter of the intercalated cations, it seems reasonable to assume that the interaction between the cations and the water travelling through the nanochannels may govern the water transport.

**Water transport mechanism through cation intercalated GOMs**

To explain the water transport mechanism through cation intercalated GOM, we calculated the slip length ($l_s$) for the cation intercalated GOMs. We refer to Supplementary Note 2 for understanding the concept of slip length[2,26,27]. In addition, a detailed explanation and calculation of slip length based on our water flux and XRD measurements are discussed in Supplementary Note 2 and Supplementary Figs. 2–4. In brief, the water flux according to the HP equation was calculated based on the membranes thickness and interlayer space. In the calculation of the water flux, the available space for water transport in

**Table 1 | Mass of intercalated cations in X-M-GO and their hydrated/ionic diameter[38]**

| Intercalated cations | Na+ | K+ | Mg2+ | Ca2+ |
|---|---|---|---|---|
| Membrane mass increase (mg) | 0.28 | 0.15 | 0.66 | 0.49 |
| Number of moles of intercalated cations (µmol) | 4.79 | 2.01 | 6.93 | 4.42 |
| Hydrated diameter-$D_H$ (Å) | 6.62 | 7.16 | 8.24 | 8.56 |
| Ionic diameter-$D_I$ (Å) | 1.90 | 2.66 | 1.30 | 1.98 |

Source data are provided as a Source Data file.

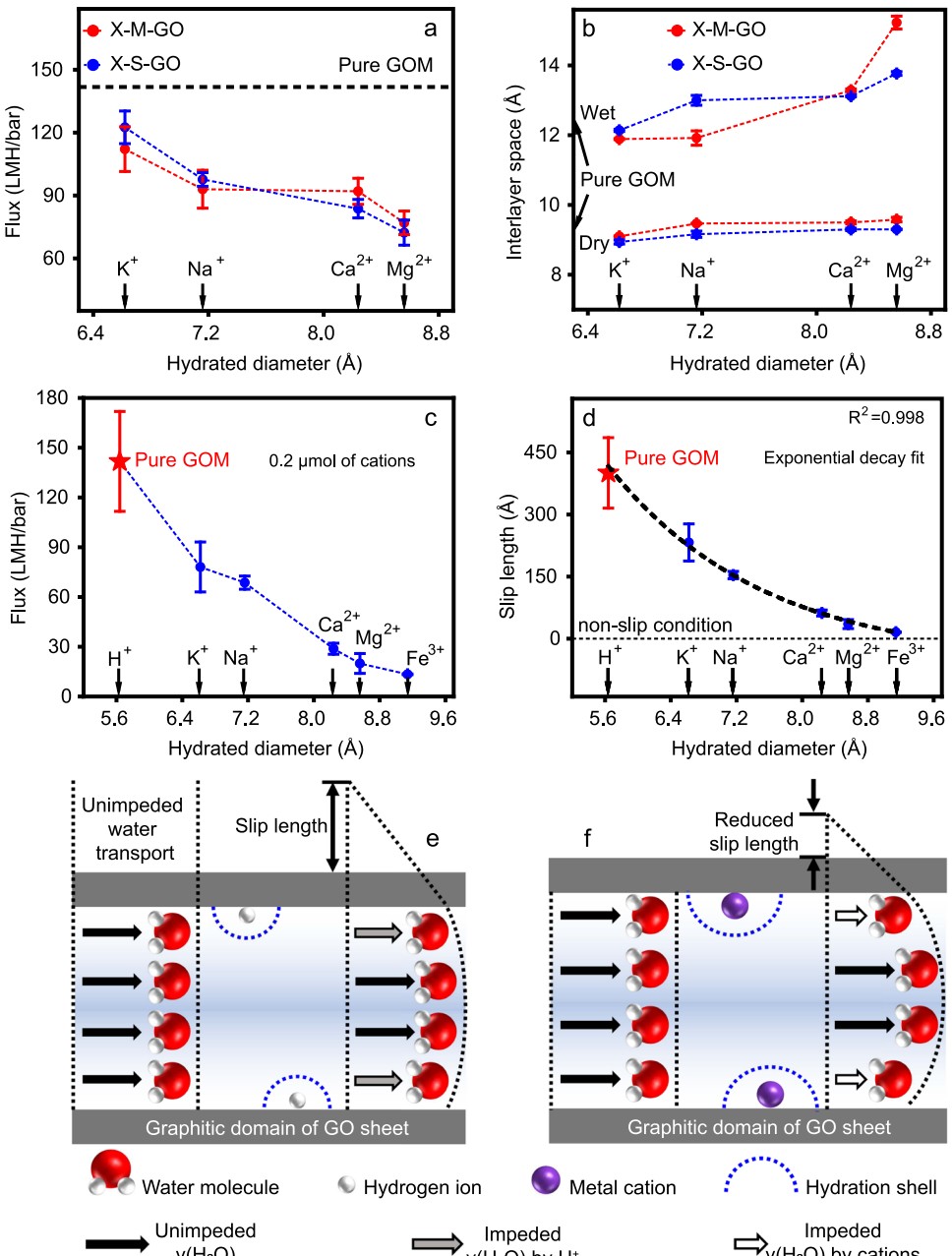

**Fig. 2 | Water transport through cation intercalated GOMs. a** Variation of water flux of cation intercalated GOMs against hydrated diameter of cations. The red scatters correspond to the GOMs prepared by membrane intercalation method (X-M-GO) and blue scatters correspond to solution intercalation method (X-S-GO). The unit of water flux is L m$^{-2}$ h$^{-1}$ bar$^{-1}$ (LMH/bar). The black dashed line represents the flux of pure GOM. **b** Interlayer space of cation intercalated GOMs against hydrated diameters. The hydrated diameters of cations are listed in Table 1. **c** Variation of water flux of X-S-GO with controlled moles of cation, where H$^+$ corresponds to the case of water flux of pure GOM. **d** Variation of slip length with hydrated diameter of cations; slip length is calculated using experimentally observed water flux and interlayer space. The dotted black line displays an exponential decay fit with a coefficient of determination ($R^2$) of 0.998. For each data point shown in (**a–d**), three separate membranes were tested. Error bars indicate the standard deviation from these three membranes. **e, f** Schematic diagrams of water transport through pure GOM (**e**) and cation intercalated GOM (**f**). v(H$_2$O) corresponds to the velocity of water molecules. The dimension of water molecules, GO sheets and hydrated cations are chosen for illustration and are not intended to represent the realistic scale. Without the influence of cations or hydrogen ion (H$^+$), the water flux through graphitic domains is illustrated as unimpeded[2]. Source data are provided as a Source Data file.

between the GO layer was estimated using the interlayer space from XRD measurements and the diameter of carbon atoms[2]. Further, we estimated that the intercalated cations may not change the average available space as the ratio of number of intercalated carbon atoms to cations is ~144 (Supplementary Note 3). With that only a limited number of cations exist in the channels causing no substantial changes to the available space for water transport. From the difference between

the experimentally measured and calculated water flux the slip length is determined[2].

The most obvious assumption that could explain a decreased water flux or changes in slip length by cation intercalation is that the cations cause steric hindrance for the transport of water molecules. If that was true, the water flux trend should follow the ionic diameter of the cations. As shown in (Supplementary Note 4 and Supplementary

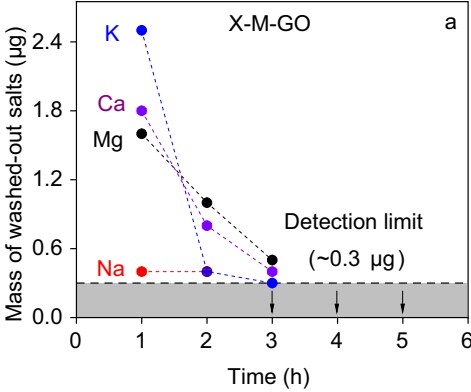

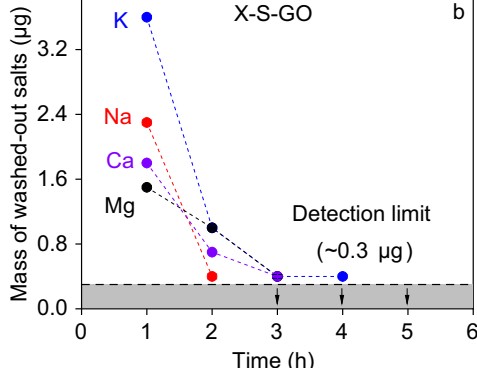

**Fig. 3 | Stability test of intercalated cations.** Number of cations in permeate over the course of washing time for (**a**) GOMs prepared by membrane intercalation method (X-M-GO) and (**b**) GOMs prepared by solution intercalation method X-S-GO. The membranes were filtered with DI water and the resulting permeate was analysed by ICP-OES to determine the mass of washed-out cations. Error bars are standard deviations from three permeate samples. Source data are provided as a Source Data file.

Fig. 5) the water transport does not follow this trend. In contrary the $K^+$ would cause the largest steric hindrance as they have the largest ionic diameter. However, the water transport for $K^+$ is the highest among tested cations. For $Mg^{2+}$, the cation with the smallest ionic diameter, shows comparably lower water transport, despite showing a larger interlayer space compared to $K^+$ intercalated GOMs. With that it can be ruled out that steric hindrance of the cations is the major contributor to the hindrance of water transport upon cation intercalation.

As shown in Fig. 2d, the slip length decreases with the increased hydrated diameter. This decrease in slip length can be well fitted with an exponential decay function. Interestingly, when considering $H^+$ as the cation intercalated in pure GOM, it also fits the exponential relation. Moreover, the exponential function decays towards zero slip-length (non-slip condition) for the highest hydrated diameters.

This allows us to draw interesting conclusions about the water transport in cation intercalated GOMs and GOMs in general. GO consists of functionalized and graphitic domains. In GOMs the water flux is considered to mainly move along the graphitic domains[2,20]. The water flux along graphene-based nanochannels is often described based on the HP equation considering a large slip length that exceeds the channel width[28,29]. The large slip length describes the low friction of water along the graphitic planes. As shown in Fig. 2e for pristine graphitic nanochannels, the slip length is large and has been reported to be around $100–1000$ Å[2]. This is in line with our calculated value ~$400$ Å (Fig. 2d) for the original GO based on our experimental result. After cation intercalation, the slip length decreases. Hence, the intercalation of cations causes a hindrance to the water transport in the nanochannels (Fig. 2f). This hindrance increases exponentially with the hydrated diameter of intercalated cations, as shown in Fig. 2d. This shows that enhanced interaction between water molecules and cation decorated graphitic nanochannel is responsible for the decrease in water flux. The fact that the hydrated diameter of $H^+$ fits the exponential function indicates that hydrated hydrogen ions may govern the water transport in the original GOM (Fig. 2e).

Moreover, the exponential decay of the slip length approximates the non-slip condition (see Supplementary Fig. 2c for visualisation of the non-slip condition, $l_s = 0$). That means that if cations highly increase the interaction between the nanochannel walls and the water, the water transport follows the HP equation. This is a valuable insight as this is experimental support for the use of the HP equation in nanochannels with slip lengths that exceed the channel width. Here it is to be noted that so far, the justification for using the HP equation relied only on molecular dynamic (MD) simulations[29] and is debated in the literature[30].

## Stability of intercalated cations

Differences in cation intercalation stability may be responsible for the water flux trend. To exclude that, a control experiment testing the stability of the intercalated ions was conducted. For that, pure DI water was fed through the membranes, and the permeate of this experiment was collected and analysed by ICP-OES to determine the mass of cations that might have been washed out. The experimental details are discussed in Supplementary Note 5, Supplementary Figs. 6, 7 and Supplementary Tables 1–3. As shown in Fig. 3 for all membranes, after 3–5 h, there were no detectable cation in the permeate. The total mass of washed-out cations is below 5.5 μg (Supplementary Table 3), which is negligible compared with the measured mass of intercalated cations (Table 1). Hence, we conclude that the cation intercalation is stable in our water flux experiments.

Since the above reason could be excluded to be responsible for the observed trend in water flux, the following examination further focuses on the interaction between intercalated cations and water molecules. As shown above, the flux declines, and slip length increases with the increase of the hydrated diameter of the intercalated cation. At the same time, the interlayer space is increased with the increase in hydrated diameter as well. A higher increase in the interlayer space upon wetting the membrane is associated with increased intercalated layers of water. We speculate that the cations with higher hydration diameter have a stronger attraction towards water inside the capillaries. To investigate that, we performed AIMD simulation with several intercalated cations and investigated the water density around the specific cation.

## Simulation study on water affinity towards cation intercalated graphitic nanochannels

Figure 4a shows a simulation cell used to investigate the behaviour of nanoconfined water in the presence of cations. The simulation cell resembles a bilayer of graphene with two cations attached to the basal plane. The bilayer configuration is achieved by employing periodic boundary conditions. The space between the layers is filled with water molecules. Graphene was chosen as a model as various studies confirm that main group metal cations intercalate mainly via cation-π interaction[31,32]. With that in GOMs, the main group metal cations would attach to the graphitic areas where π electrons exist. The cations may also attach to functional groups (see the simulation cell in Supplementary Fig. 8a). Hence, simulations were conducted with functionalized graphene sheets as the substrate for cation intercalation (see Supplementary Note 6). For both cases, graphitic and functionalized nanochannels, we examined the rearrangement of water molecules by projecting their distribution on the graphitic/functionalise

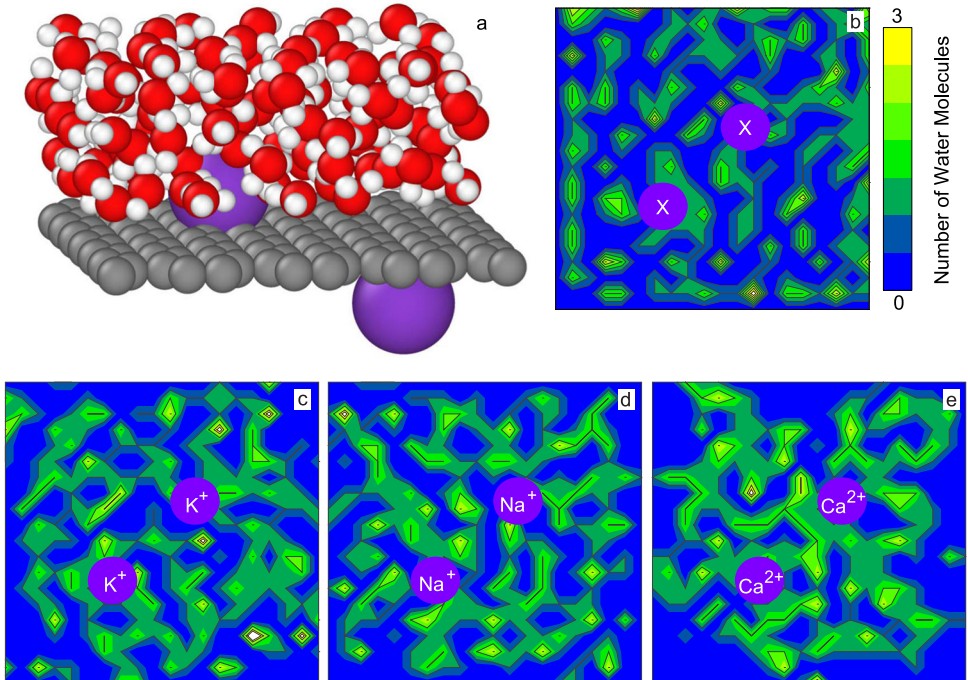

**Fig. 4 | Simulation study on the water affinity towards cation intercalated graphitic nanochannels in a model bilayer graphene. a** Schematic diagram of the simulation cell at initial state (0 ps) in ab initio MD simulation. The supercell consists of a graphitic sheet (grey spheres) intercalated with two cations (violet spheres) and filled with water molecules (red and white spheres are oxygen and hydrogen atoms, respectively). The periodic boundary conditions ensure the nanoconfinement. The dimension of the simulation cell is x = 17.23 Å, y = 17.05 Å and z = 14 Å. **b** Distribution of water molecules at the initial state. Violet circles are the position of cations. x is a placeholder for different types of cations. The number of accumulated water molecules is represented by the colour range in the scale. **c–e** Distribution of water molecules after 4 ps for graphitic sheets intercalated with K$^+$, Na$^+$ and Ca$^{2+}$ ions, respectively. Source data are provided as a Source Data file.

domain plane. Starting with randomly distributed water molecules around the ions (Fig. 4b and Supplementary Fig. 8b), we let the molecules relax over 4 ps. After that period, the water molecules rearranged more densely around the cations (Fig. 4c–e and Supplementary Fig. 8c–e). We tried relaxing Mg atoms on graphene; however, the Mg atoms did not bind with the graphene sheet and stabilised far (>4 Angstrom) from the graphene plane. As such, we did not include Mg atoms in our study.

In the case of functionalized channels, the water molecules rearrange during the relaxation period by getting attracted to the intercalated cations. However, no observable trend in water affinities with different cations is found (Supplementary Fig. 8c–e). Interestingly, in the case of graphitic channels the water molecules show a higher density around the Ca$^{2+}$ than the monovalent cations of K$^+$ and Na$^+$ (Fig. 4c–e). The higher density around the Ca$^{2+}$ suggests that cations with larger hydration shells attract water more tightly in the confined environment of graphene nanocapillaries. Despite allowing a larger number of water molecules to intercalate into the capillaries and expanding the space for the water to travel, the flux declines and slip length decreases, as shown in our experimental results. From the simulation results, we may conclude that this decline is based on the increased interaction between water molecules and intercalated cations. As the differences in water affinities of cations in the simulations was prominent in the case of cation-π interaction and not observable in cation intercalation with functionalized areas, it suggests that the cation-π interaction on the graphitic areas of GO is the more dominant case for governing the slip length. A similar increase in friction was predicted with AIMD simulations of water transport with impurities in the graphitic nanochannel walls[10]. In this case, the friction was correlated with the hydrogen bond ability of impurities. This increase in interaction between cations and water molecules may affect the slip length in two different ways. The slip length can be interpreted as a description of the friction of water with the nanochannel walls. Increased attraction of water molecules via the absorbed cation would cause increased friction and explains the trend of decreasing slip length with increasing hydrated diameter. The large slip length in graphene-based nanochannels may also be explained by a depletion layer with reduced viscosity close to the channel walls[26]. Increased interaction between the channel walls and the intercalated cations may also reduce the depletion layer and increase the viscosity close to the channel walls, explaining the decrease in slip length with increasing hydration diameter.

## Discussion

We present a study on water transport through cation intercalated GOMs. With that, experimental access to the role of impurities with different water affinities in graphene-based nanochannel is gained. We found that the flux of cation intercalated GOMs was decreased with the increase of interlayer space. We explain this by showing that the slip length of the cation intercalated GOMs scales exponentially with the hydrated diameter of the respective cation. This allows us to hypothesise that the water flux in original GOMs is influenced by hydrated hydrogen, which is omnipresent in aqueous environments. Our simulation study further supports that the hindrance of water transport through the cation decorated nanochannels is due to increased interaction between the water molecules and the nanochannel. Further, the non-slip conditions for high hydrated diameters offer experimental support for using the HP equation in graphene-based nanochannels. Hence, our study suggests that the water transport in graphene-based nanochannels follows the HP equation. The slip length may change dependent on the water affinity of impurities. As we showed, in the case of cation intercalation, the slip length decays exponentially with the hydrated diameter (i.e. water affinity). Future studies may find a way to further explore and confirm the exponential relationship between hydrated cations and GOM slip length. With that, our study may offer a starting point towards a deeper understanding of

the water transport through graphene-based nanochannels with water-attracting impurities. In addition, these may pose valuable insights for future applications where graphene-based nanochannels in aqueous environments play a promising role.

## Methods

### GOM preparation
GO powder containing single-layered laminates with a lateral size of ~500 nm (Jiangsu XFNANO Materials Tech. Co. Ltd) was dispersed in deionized water at the concentration of ~0.1 mg/mL. The GO suspension was then ultrasonicated for 2 h under ambient conditions, avoiding GO reduction at elevated temperatures. GOMs were fabricated onto polyvinylidene fluoride (PVDF, 0.1 µm pore size) substrate by filtering 5 mL of GO suspension under vacuum at ~0.9 bar pressure, and the obtained GOMs were air-dried for 24 h.

### Membrane intercalation (X-M-GO)
GOMs were separately soaked in 10 mL of 10 mM salt solutions using NaCl, KCl, $MgCl_2$ and $CaCl_2$ for 24 h, followed by air-drying for 24 h. The average mass increase of these GOMs was measured as shown in Table 1.

### Solution intercalation (X-S-GO)
NaCl, KCl, $MgCl_2$ and $CaCl_2$ powders with the number of moles shown in Table 1 were mixed with 5 mL, ~0.1 mg/mL GO solution to prepare cation intercalated GO solutions. The dispersibility of the mixture was ensured by incubating the suspension for 24 h. Then, the suspension was filtered through PVDF substrate under ~0.9 bar pressure. Another set of intercalated GOMs was prepared from 5 mL, 0.1 mg/mL of GO solution containing 0.2 µmol of $K^+$, $Na^+$, $Ca^{2+}$, $Mg^{2+}$ and $Fe^{3+}$.

### Characterisation of GO sheets and GOMs
Transmission electron microscopy (TEM, JEOL, JEM-2100F, Japan) was used to investigate the dimension of GO sheets on lacey carbon grid. The cross-sectional structure and the thickness of pure GOM on PVDF substrate were characterised by field emission scanning electron microscopy (FE-SEM, Hitachi S-4800, Japan). The X-ray diffraction (XRD, Bruker D8 focus, Germany) patterns of GOMs in both dry and wet (soaked in deionized water for 10 min) conditions were obtained using Cu-Kα radiation (λ = 1.54 Å). The chemical compositions of GOMs were analysed by X-ray photoelectron spectroscopy (XPS, Thermo Scientific Al-Kα radiation, USA).

### Membrane permeation and separation tests
Membrane permeation tests were carried out by vacuum filtration. The cation intercalated GOMs on PVDF substrates were tightly clamped between feed and permeate compartments. Deionized water was permeated through GOM for 5 h, and the cations concentration in permeate and water volumes were recorded every 1 h. The cation concentration was measured by Inductive Coupled Plasma-Optical Emission Spectrometer (ICP-OES, Varian 710-OES, USA). The flux of cation intercalated GOM was calculated by the following equation:

$$J = \frac{\triangle V}{AtP} \tag{1}$$

where $J$ is flux in unit of L/m²/h/bar (LMH/bar), $\triangle V$ is the total volume of solution in permeate side, $A$ is the effective area of GOM ($A = 1.77$ cm²), $t$ is the permeation time ($t = 5$ h), and $P$ is the applied pressure in vacuum filtration ($P \approx 0.9$ bar).

### Computational details
Molecular dynamic (MD) simulations have been performed using the CP2K suite of programs[33]. A double-zeta-valence plus polarisation MOLOPT basis set[34] is used to expand the Kohn-Sham orbitals, while the electronic structure is described within the Gaussian and Plane Waves framework[35]. Exchange and correlation are considered through the Perdew-Burke-Ernzerhof[36] functional, while Goedecker type (GTH) pseudopotentials[37] have been used to model the interaction of valence electrons with the atomic cores.

The metals absorbed on the graphene/GO structures were first relaxed using DFT through the Quickstep[35] module of CP2K, until forces were inferior to 1.0E−03 a.u. Later, ab initio MD simulations, based on the propagation of the equations of motion with Langevin dynamics, were performed in the canonical ensemble at 310 K. The graphene/GO structures have been kept fixed, while water has been allowed to move freely. Each simulation ran for 4 ps, using a time-step of 1 fs.

## Data availability
Source data are provided with this paper. The plot data generated in this study have been deposited in the Figshare as https://doi.org/10.6084/m9.figshare.20847769 Source data are provided with this paper.

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

## Acknowledgements

The authors acknowledge the support from Prof. Wei Guo of the Technical Institute of Physics & Chemistry, Chinese Academy of Science. T.F. acknowledges the UNSW Scientia Scholarship. Priyank Kumar acknowledges the UNSW Scientia Fellowship and the ARC for financial support through the Discovery Early Career Researcher Award (DE210101259).

## Author contributions

X.W., T.F. and R.J. conceived the project and methodology. X.W. performed the experiments. The data were analysed by X.W. and T.F. with the help of X.J. and R.J. T.M., and P.K. performed computational simulations. X.W. prepared the visualisation. X.W., T.F. and R.J. wrote the paper. All authors contributed to the discussions. R.J. supervised the project.

## Competing interests

The authors declare no competing interests.
