## [Peer Review File · Nature Communications]

REVIEWER COMMENTS

Reviewer #1 (Remarks to the Author):

Some of my principal criticism is that, while the paper is well written and the details provided help understand the physics, the findings are neither new nor surprising. It has been well known that it is the solvation shells surrounding ions and the formation of hydrogen bonds with oxygenated moieties which cause the most pronounced deviations from the unimpeded flow of water through graphite and graphene sheets and nanotubes. The decoration of graphite with any type of associating/solvatable impurity will fundamentally change its surface adsorption (see papers by K. Gubbins , e.g. Muller, E.A., et. al. Adsorption of water on activated carbons: A molecular simulation study. J. Phys. Chem. 100, 1189–1196 (1996)) and the flow characteristics (see papers by P. Carbone, e.g. Abraham, J. et al. Tunable sieving of ions using graphene oxide membranes. Nature Nanotechnology 1–6 (2017)). The changes are influenced by the discrepancy between the very weak solid-fluid interaction between carbon and water and the extremely strong water/associating site or water/ion interaction which effectively binds water and creates “nucleation” sites which grow (in the case of adsorption) or act as imperfections in the smooth walls impeding flow.

The paper is a very good example of incremental work in the field, leading to the supporting the above view. However, as mentioned before, does not provide any new insights which were not known and expected.

Reviewer #2 (Remarks to the Author):

This manuscript reports the correlation between water transport rate and channel wall interactions in cation-modified graphene oxide nanochannels. The authors found the water flux decreased with increased interlayer space of the nanochannel and explained this using ‘slip length’ change of the membrane channels. While the study is interesting, I have a major concern on this as the increased interlayer space is not the exact free space for water transport. There should be a certain number of ions existed in the channel, which may act as steric hindrance. The authors should discuss about this before making assumptions on the effects of channel wall interactions or slip lengths. This work is not recommended for acceptance.

In addition, I have following concerns:

1. Friction-free movement of water occurs in graphitic nanochannels; However, there are abundant oxidized regions in GO nanochannels, where the water transport should NOT be friction-free.
2. The Introduction emphasized the graphitic nanochannels with impurities. What kind of impurities was referred here?
3. How was the accuracy for the determination of weight gain to calculate the molarity of intercalated ions in GO membranes?
4. Line 110-112: Mixing salt solution with GO solution for same molar concentrations is reasonable to do, but it remains questionable whether the resulting membranes contain same molar concentration of ions. Confirming these membranes containing same amount of ions is absolutely an prerequisite here before approaching any conclusions from the related experimental results.
5. There is an undefined reference in Section 2 of supporting information.
6. Line 191: It seems Figure S4 is not for the visualization of the non-slip condition.
7. For the simulation, at least a bilayer of graphene should be constructed for modelling to give better match with the authors’ claim.
8. And considering GO was used for experiments, GO rather than graphene should be used in the simulation model. In this case, metals ions will be favorably attached to different regions of GO. For example, divalent ions may have larger possibility to locate at the carboxyl region of GO. Therefore, the simulation model presented here is hardly convincing.
9. The expanded interlayer distance from XRD results may not be the actual free space for water transport. While I believe the interactions between water and intercalated ion should partially account for the declined flux, the intercalated ions may also play as steric hindrance to slow down

the transport.

RESPONSE TO REFEREES

Reviewer #1 (Remarks to the Author):

Comments: Some of my principal criticism is that, while the paper is well written and the details provided help understand the physics, the findings are neither new nor surprising. It has been well known that it is the solvation shells surrounding ions and the formation of hydrogen bonds with oxygenated moieties which cause the most pronounced deviations from the unimpeded flow of water through graphite and graphene sheets and nanotubes. The decoration of graphite with any type of associating/solvatable impurity will fundamentally change its surface adsorption (see papers by K. Gubbins, e.g., Muller, E.A., et. al. Adsorption of water on activated carbons: A molecular simulation study. *J. Phys. Chem.* 100, 1189–1196 (1996)) and the flow characteristics (see papers by P. Carbone, e.g., Abraham, J. et al. Tunable sieving of ions using graphene oxide membranes. *Nature Nanotechnology* 1–6 (2017)). The changes are influenced by the discrepancy between the very weak solid-fluid interaction between carbon and water and the extremely strong water/associating site or water/ion interaction which effectively binds water and creates “nucleation” sites which grow (in the case of adsorption) or act as imperfections in the smooth walls impeding flow.

The paper is a very good example of incremental work in the field, leading to the supporting the above view. However, as mentioned before, does not provide any new insights which were not known and expected.

Response: We thank the referee for the positive aspects of the comment as well as the constructive criticism. We agree with the referee that our paper **supports the findings from simulations** that water-binding sites such as impurities or defects have an important role in water transport. This is well investigated and understood from a **theoretical/computational point of view**. One example is first article suggested by the referee (Muller et. Al, *J. Phys. Chem.* 100, 1189–1196 (1996)). We have discussed this article in our revised manuscript. However, **experimental observation** is still **critically missing** in the literature. **Our work is the first article that reports the experimental evidence of this phenomenon.**

The second article suggested by the referee in which Abraham et.al (experimental work) focussed on rejecting hydrated ions by controlling the interlayer space of GO membranes. The aspect discussed by them, related to water transport, is limited to the effect of the interlayer space, and they observed with **decreased interlayer space, less amount of water can pass through the membranes**. This is intuitively understandable. However, our study shows that a larger interlayer space **does not necessarily imply a higher water flux**. We show that even a limited number of cations intercalated with GO (1 cation for every 144 carbon atoms) drastically influences water transport. By varying the species of cations with different hydrated diameters,

we can show that the water transport depends exponentially on the hydration shell of intercalated cations.

This is a new finding that has not been experimentally investigated. To the best of our knowledge, this is the first study that gains experimental access to the role of water adsorption sites in such 2D laminar membranes. Our study is not incremental work but a critical one to support findings from simulations with reliable and repeatable experimental observation.

Reviewer #2 (Remarks to the Author):

Comments: This manuscript reports the correlation between water transport rate and channel wall interactions in cation-modified graphene oxide nanochannels. The authors found the water flux decreased with increased interlayer space of the nanochannel and explained this using 'slip length' change of the membrane channels. While the study is interesting, I have a major concern on this as the increased interlayer space is not the exact free space for water transport. There should be a certain number of ions existed in the channel, which may act as steric hindrance. The authors should discuss about this before making assumptions on the effects of channel wall interactions or slip lengths. This work is not recommended for acceptance.

Response: We thank the referee for this insightful criticism. The referee pointed out that intercalated cations may 1) decrease the available space for mass transport and 2) cause steric hindrance and suggested both phenomena can contribute to the decreased water flux. However, with an extensive analysis and calculation, we can confirm that both cannot be the main reason for decreased water flux. We explain this one by one (**please also see the response to the last comment**):

1) Role of available space in decreased water flux:

The calculation of available space is shown in SI, section 2.1. The available space here is the distance between the electron clouds of adjacent graphene planes. As calculated, a small number of cations are intercalated compared to the host carbon atoms (carbon/cation ~144), suggesting a negligible impact on available space by cations for water transport.

2) Role of steric hindrance in decreased water flux:

The steric hindrance is related to the ionic radius, as the larger ionic radius of cations results in a higher steric hindrance, which in other words, can be the physical presence of the cations between the free space of GO layers. According to the referee, the water flux decrease should follow the trend of steric hindrance caused by the cations, i.e. higher steric hindrance results in **lower water flux**. To understand this, we investigated the relationship between ionic diameter and water flux/slip length. As described in SI, section 2.3, the highest steric hindrance (largest diameter) cation K^+ showed the **highest flux** and slip length. With this, we can rule out that the water flux

depends on the steric hindrance caused by the intercalated ions.

In summary: The water flux is neither determined by the available space for water transport nor the steric hindrance of the intercalated cations but only by the hydration shell of cations or the water affinity of the intercalated cations. As explained in our study, the water flux and slip length of the GOMs depends on the hydrated diameter of intercalated cations in an exponential decay trend.

Comment: In addition, I have following concerns:

1. Friction-free movement of water occurs in graphitic nanochannels; However, there are abundant oxidized regions in GO nanochannels, where the water transport should NOT be friction-free.

We agree that the water transport in GO nanochannels is not entirely friction-free. However, the large slip length of GO membranes suggests that the water transport experiences low friction. This is still controversially (Chong et al. Journal of Membrane Science 549, 385–392 (2018) as ref 28 in manuscript, Nair et al. Science, 335, 442–444 (2012) as ref 2 in manuscript) discussed in the community as the calculation of slip length is based on the assumption that the water follows the tortuous pathway of the GO network, and the HP equation is applicable in nanochannels with space for only a few layers of water. Our study shows that by increasing the friction inside the nanochannels, the slip length decays towards zero and hence supports the use of the HP equation and the assumption about the pathway of the water molecules. We added more discussion in the Introduction part on page 2, lines 29-31 and 40-42.

2. The Introduction emphasized the graphitic nanochannels with impurities. What kind of impurities was referred here?

The impurities here are the intercalated cations, which have different water affinities. These impurities enable us to tune the structure of nanochannels in GO and understand their effect on water transport. We also add more description about this in the Introduction part on page 3, lines 85-88.

3. How was the accuracy for the determination of weight gain to calculate the molarity of intercalated ions in GO membranes?

We thank the referee offers the opportunity to clarify this. The use of the weight gain method to calculate the mass of intercalated ions was first introduced by Chen et al. (Nature 550, 380–383 (2017) as ref 14 in manuscript) in their landmark article on cation intercalation. We also used a similar method to control the molarity of intercalated cation. The description and reference are added on page 5, lines 115-118.

4. Line 110-112: Mixing salt solution with GO solution for same molar concentrations is reasonable to do, but it remains questionable whether the resulting membranes contain same molar concentration of ions. Confirming these membranes containing same amount of ions is absolutely an prerequisite here before approaching any conclusions from the related experimental results.

We would like to clarify that weight gain as a method to determine the number of intercalated ions was **only used in the specific case of intercalating cations via soaking the membrane in the salt solution**. Though widely used, this method offers errors. Thus, for calculating the slip length, we chose to intercalate the ion via mixing salt solution on GO solution prior to fabrication of the membranes. We then measured effluent concentration with a more accurate ICP-OES method to ensure that the cations were intercalated and not washed out. We could confirm that 99.99 % of the cations remain in the membrane. This is discussed on page 5 lines 126-130

5. There is an undefined reference in Section 2 of supporting information.

We have added the reference in Section 2 of the supporting information.

6. Line 191: It seems Figure S4 is not for the visualization of the non-slip condition.

The visualization of non-slip conditions is shown in supporting information in Figure S2C. We have modified the text on page 9, line 228.

7. For the simulation, at least a bilayer of graphene should be constructed for modelling to give better match with the authors' claim.

The simulation cell used in this study consists of a bilayer of graphene and two cations. These two cations were attached to the basal plane of graphene. We further employed periodic boundary conditions of this bilayer configuration. More description of simulation cell construction is added on page 11, lines 263-265.

8. And considering GO was used for experiments, GO rather than graphene should be used in the simulation model. In this case, metals ions will be favourably attached to different regions of GO. For example, divalent ions may have larger possibility to locate at the carboxyl region of GO. Therefore, the simulation model presented here is hardly convincing.

Based on previous experimental and simulation studies (ACS Nano 2014, 8, 1, 850–859, ref 29 in the manuscript. ACS Nano 2013, 7, 1, 428–437, ref 30 in the manuscript.), it can be understood that the metal cations intercalate within GO via cation- π interaction. As this interaction is only possible on the graphitic domains, we chose to simulate a graphene bilayer where π electrons exist. This discussion is also added on page 11, lines 265-268.

9. The expanded interlayer distance from XRD results may not be the actual free space for water transport. While I believe the interactions between water and intercalated ion should partially account for the declined flux, the intercalated ions may also play as steric hindrance to slow down the transport.

As suggested by the referee, we investigated the actual free space for water transport. Based on our carbon/cation ratio calculation of 144, we can say that only a limited number of cations exist in the nanochannels, which are insufficient for a substantial change in the available space for water transport. The detailed explanation is added

on page 8, lines 188-197 in the revised manuscript.

As stated above, we have investigated the role of steric hindrance in water transport on page 8, lines 198-207 of the revised manuscript. *Steric hindrance is related to the ionic diameter of cations, where larger cations have higher steric hindrance.* To have a clear picture, we have plotted the water flux and slip length against the ionic diameter of intercalated cations (see below and shown in Figure S5 in SI). We found that neither the water flux of intercalated GOMs nor the slip length followed the trend of ionic diameter variation, suggesting that the steric hindrance is not the major contributor to determining the water transport in cation intercalated GOMs.

Figure S5: Steric hindrance affecting the water transport through 0.2 μmol of cation intercalated GOMs. (A) Variation of water flux of cation intercalated X-S-GOMs against ionic diameter of cations. (B) Variation of slip length with ionic diameter of cations. Error bars indicate the standard deviation from three tested membranes.

REVIEWER COMMENTS

Reviewer #2 (Remarks to the Author):

The authors have addressed most of my concerns on their previous manuscript. But I am still concerned about the interactions responsible for the intercalation of cations in GO. While it is true that cation- π interaction only occurs between cations and graphitic areas of GO, the authors also need to clarify the cation complexing by sp^3 carbon areas of GO. Will such complexing affect the slip length that the authors investigated? Is it possible to perform another simulation using complexed cation and GO model?

RESPONSE TO REFEREES

Reviewer #2 (Remarks to the Author):

The authors have addressed most of my concerns on their previous manuscript. But I am still concerned about the interactions responsible for the intercalation of cations in GO. While it is true that cation- π interaction only occurs between cations and graphitic areas of GO, the authors also need to clarify the cation complexing by sp^3 carbon areas of GO. Will such complexing affect the slip length that the authors investigated? Is it possible to perform another simulation using complexed cation and GO model?

Response: We thank the referee for the positive comments and valuable suggestions. We truly believe that addressing the suggested comments will strengthen our analysis. During our initial simulations, we also thought that the sp^3 carbon might offer intercalation sites for the cations in graphene oxide, and we have been continuously studying that phenomenon. We conducted simulations with functionalized graphene to understand the water affinity of cations intercalated at sp^3 carbon sites. The referee rightly pointed out that the role of functionalized areas should also be investigated; hence, we have included additional simulation results in the revised manuscript. Our new results further support that the graphitic sites are dominant in governing the change in slip length, as explained below.

For the sp^3 carbon areas of GO, the cation intercalation possibly occurs at oxygen functional groups such as hydroxy, epoxy and carboxy groups. In graphene oxide, the carboxy groups are located at the edges of the flakes with limited effect on the water transport through graphitic nanochannels (Reference 8 of SI). This suggests that the slip length will not change due to the interaction at the carboxy group of GO. We conducted simulations for cation intercalation on hydroxy groups (see **Figure S8 in SI, section 4**). The rearrangement of water molecules was investigated after relaxing the cell for 4 ps (see **Figure S8C-E in SI, section 4**). This is similar to the simulation study we conducted on the graphitic area of GO (**Figure 4**). We found that Na^+ , K^+ and Ca^{2+} ions showed no noticeable trend in water affinity if attached to functionalized areas of graphene oxide. Based on that, we believe that the cation intercalation with sp^3 carbon area cannot explain the trend of the slip length observed in **Figure 2D** of the manuscript. The water affinity of intercalated cations only shows a matching trend when the cations interact with the sp^2 carbon (**Figure 4C-E in manuscript**). This suggests that the cations intercalated at the graphitic areas are more dominant in governing the slip length. We have added this discussion in the manuscript on pages **11-12, lines 274-280, 284-288 and 294-297, as well as in the SI section 4 on pages 8-9, including figure S8**.

REVIEWERS' COMMENTS

Reviewer #2 (Remarks to the Author):

The author's reply is satisfactory, and the paper can be accepted now.

RESPONSE TO REFEREES

Reviewer #2 (Remarks to the Author): The author's reply is satisfactory, and the paper can be accepted now.

Response: We thank the referee.